# PROCEEDINGS A

nuclear physics

pear shapes of atomic nuclei, octupole collectivity, Coulomb excitation, radioactive beams, electric-dipole moments

**Author for correspondence:**
P. A. Butler
e-mail: peter.butler@liverpool.ac.uk

An invited Review to mark the election of the author to the fellowship of the Royal Society in 2019.

# Pear-shaped atomic nuclei

## P. A. Butler

Oliver Lodge Laboratory, University of Liverpool, Liverpool L69 7ZE, UK

 PAB, 0000-0001-6080-9205

This review presents the current status of experimental evidence for the occurrence of reflection-asymmetric or 'pear' shapes in atomic nuclei, which arises from the presence of strong octupole correlations in the nucleon–nucleon interactions. The behaviour of energy levels and electric octupole transition moments is reviewed, with particular emphasis on recent measurements. The relevance of nuclear pear shapes to measurements of fundamental interactions is also discussed.

## 1. Introduction

The atomic nucleus is a many-body quantum system, and its shape is determined by the number of nucleons present in the nucleus and the interactions between them. As with other many-body systems spontaneous symmetry breaking will cause the shape to be distorted from spherical symmetry and the nucleus to be deformed. For axially symmetric nuclei, the nuclear shape can be parametrized in terms of a spherical harmonic (multipole) expansion. The spheroidal nuclear surface is defined by means of standard deformation parameters $\beta_\lambda$ describing the length of the radius vector pointing from the origin to the surface,

$$R(\Omega) = c(\alpha)R_0 \left[ 1 + \sum_{\lambda=2}^{\lambda_{max}} \beta_\lambda Y_{\lambda 0}^*(\Omega) \right], \qquad (1.1)$$

with $c(\alpha)$ being determined from the volume-conservation condition and $R_0 = r_0 A^{1/3}$ for a nucleus of mass $A$.

The simplest shape distortion is *quadrupole* deformation ($\lambda = 2$) with axial and reflection symmetry, in which case the nucleus is shaped like a rugby ball (prolate deformation, $\beta_2 > 0$). The microscopic origin of nuclear deformation arises from the neutron–proton quadrupole interaction, which will allow the nucleus to lower its total energy through quadrupole deformation (for a detailed discussion see [1]). Excited quantum states arise

**Figure 1.** Illustration of a reflection asymmetric nucleus and its coordinate system. The rotation (**R**) vector is along the $x$-axis, orthogonal to the rotating body's symmetry ($z$-) axis. If the shape is not rigid then the nucleus can vibrate between this shape and its mirror image, allowing the octupole phonon vector to align with **R** so that $I_- = R + 3\hbar$ and $\Delta i_x = 3\hbar$ (see text). From [7]. (Online version in colour.)

if this shape is rotated around an axis perpendicular to the body-fixed symmetry axis ($z$-axis). An extreme example of such behaviour is that of $^{152}$Dy, whose shape under certain conditions has a ratio of major ($z$-axis) to minor ($x$-, $y$-) axes of $2:1$ [2]. Vibrational quadrupole excitations can also arise, for example, if the shape is displaced along the symmetry axis from its equilibrium position, although the evidence for such $\beta$-vibrations has been questioned recently; see [3,4]. More complicated quadrupole degrees of freedom can arise if the shape is no longer axially symmetric but convincing evidence for triaxial systems is restricted to very few cases [5]. For a review of generalized *spontaneous symmetry breaking* in nuclei, see [6].

This review will examine the evidence for *octupole* distortion where expression (1.1) is expanded to terms containing both $\lambda = 2$ and $\lambda = 3$. In the simplest case axial symmetry is retained, but, because $\beta_3$ is non-zero, the nucleus loses reflection symmetry about the $x$–$y$-plane that passes through the origin (figure 1). It will assume a 'pear shape' in the intrinsic frame, either in a dynamic way (octupole vibrations) or by having a static shape (permanent octupole deformation). For earlier reviews on this topic, see [8–12].

## 2. Experimental evidence: rotating pear shapes

Microscopically the nucleus can lower its energy through octupole interactions, which can have a significant effect for certain combinations of proton number $Z$ and neutron number $N$. In a mean-field description of the nucleus, octupole correlations depend on the matrix elements between single particle states with $\Delta j = \Delta \ell = 3$, where $j$ and $\ell$ are the total and orbital angular momenta of the particles, respectively. Such states approach each other and the Fermi surface when either $Z$ or $N \approx 34, 56, 88, 134$, that is, at values just greater than the magic numbers where nuclei are nearly spherical. It is therefore instructive to examine the behaviour of the low-lying states in nuclei with these values of $Z$ and $N$. An important indicator of reflection asymmetry arising from strong octupole correlations in even–even nuclei is the occurrence of a rotational band having the sequence of states whose angular momenta ('spin') and parities are $I^\pi = 1^-, 3^-, 5^-, \ldots$

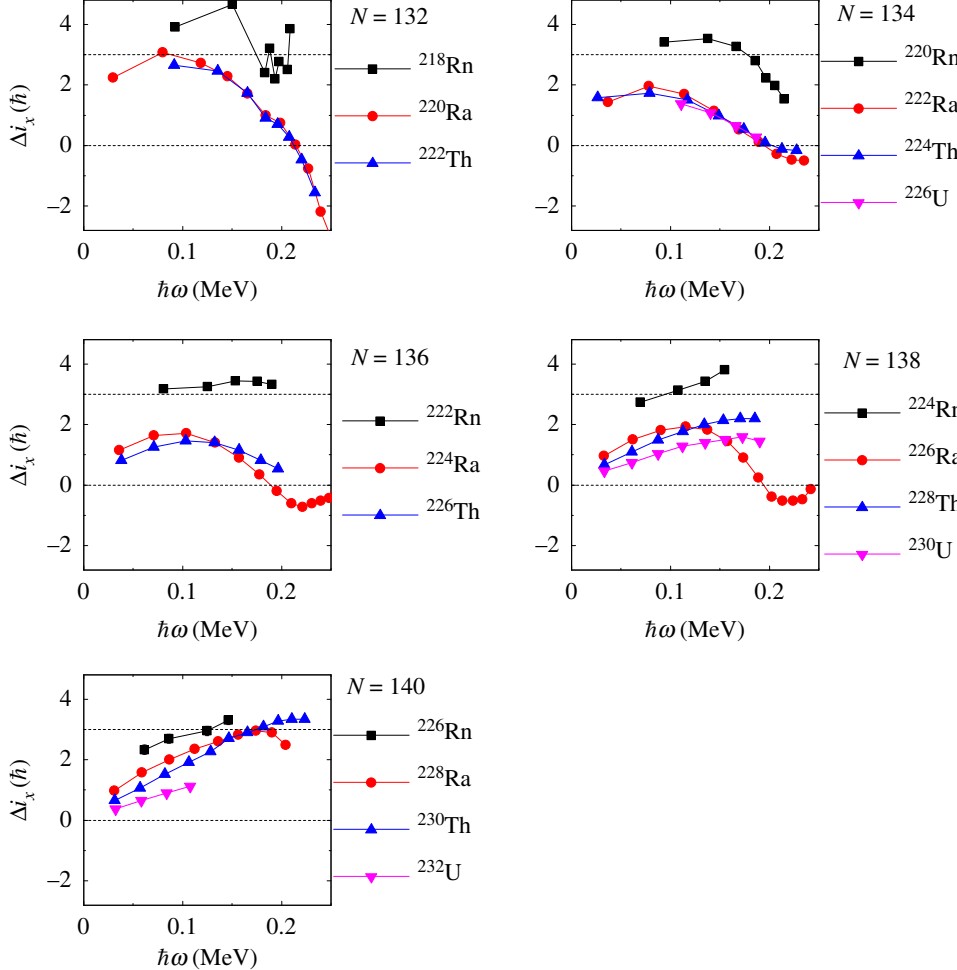

**Figure 2.** Plot of the difference in aligned angular momentum, $\Delta i_x$, against rotational frequency, $\omega$, for isotopes of Rn, Ra, Th and U (see text for explanation). For the source of the data shown here, see figs. 7,8 in [12], and see [17–19]. (Online version in colour.)

lying close in energy to the ground-state rotational band whose sequence is $I^\pi = 0^+, 2^+, 4^+, \ldots$. Such sequences of negative and positive parity states originate from instability in the octupole degree of freedom in a number of ways. One possibility is that the nucleus has permanent octupole deformation. In this case the angular momenta of the positive and negative parity states, respectively $I_+$, $I_-$, are equal to $R$, the rotation vector (which by convention is aligned to the $x$-axis). The negative and positive parity states will be interleaved and their energies are approximately given by those of a quantum rotor having moment of inertia $\mathcal{J}$, $(\hbar^2/2\mathcal{J})I(I+1)$. Another possibility is that the negative parity band arises from octupole vibrations of the rotating (quadrupole) deformed system. Here the angular momenta of the negative parity states arise from coupling that of the octupole phonon, $J$, whose magnitude is $3\hbar$, to $R$, so that $I_- = R + J$.

The study of the rotational behaviour of reflection-asymmetric nuclei has been hampered by their inaccessibility using conventional spectroscopic techniques. Nevertheless, the development of highly efficient germanium detector arrays (for review, see [13]) has revealed cascades of $\gamma$-rays de-exciting quantum states in nuclei weakly populated by spontaneous fission [14] and multi-nucleon transfer reactions [15], in rare compound nucleus channels [16], and in inelastically scattered radioactive nuclei (see next section) [7,17]. The outcome of these studies for the $Z \approx 88$,

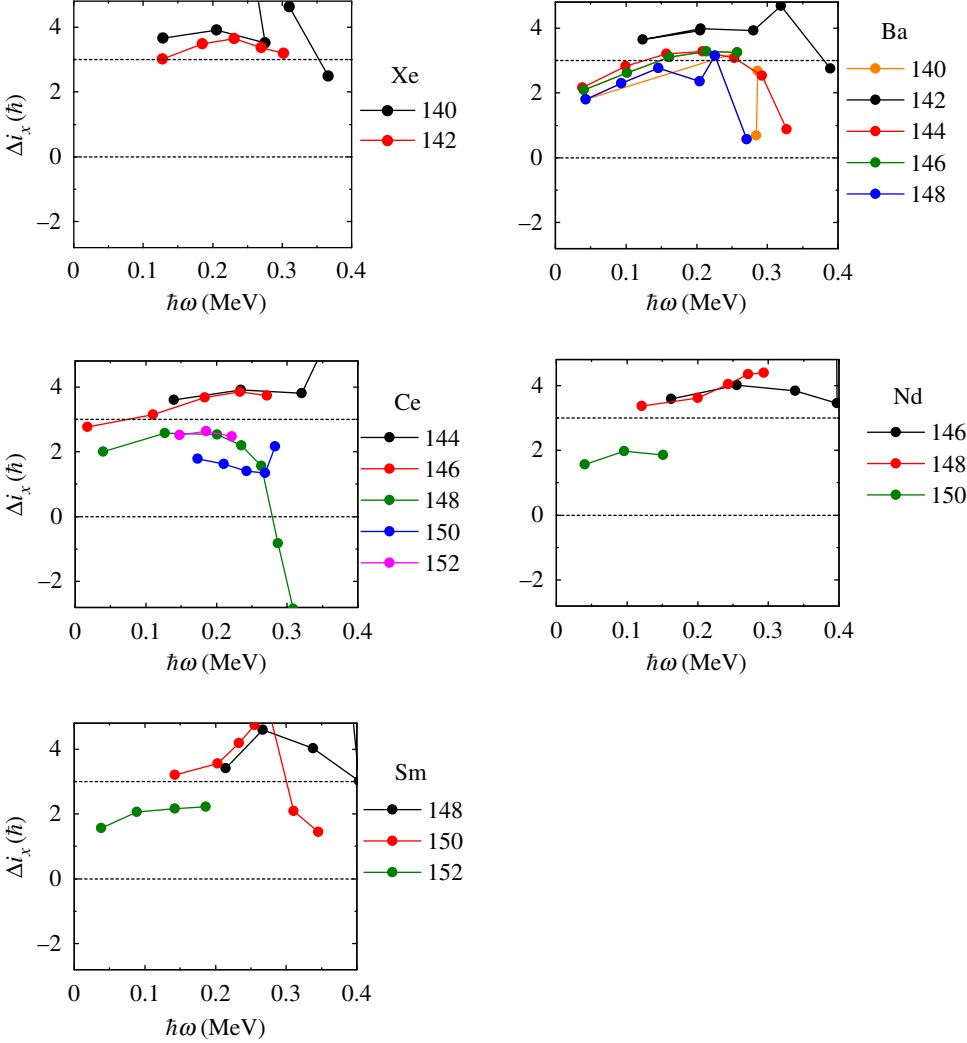

**Figure 3.** Plot of the difference in aligned angular momentum, $\Delta i_x$, against rotational frequency, $\omega$, for isotopes of Xe, Ba, Ce, Nd and Sm (see text for explanation). For the source of the data shown here, see fig. 9 in [12], superseded by this figure. (Online version in colour.)

$N \approx 134$ mass region is shown in figure 2, and for nuclei in the $Z \approx 56$, $N \approx 88$ mass region in figure 3. In these figures, $\Delta i_x = I_- - I_+$ and the rotational frequency $\hbar\omega \approx (E_I - E_{I-2})/2$, where $E_I$ is the energy of the state with spin $I$. In order to calculate $\Delta i_x$, $I_+$ is determined from its smooth variation with $\omega$ so that the corresponding value of $\omega$ is the same as that for $I_-$ [20]. For a nucleus with stable octupole deformation $I_+ = I_- = R$ at the same value of $\omega$ and the value of $\Delta i_x$ is expected to be zero. Deviations from this ideal behaviour will occur for low-spin states where nucleonic pairing will try to maintain a reflection-symmetric shape [21]. For octupole-vibrational nuclei $I_- = R + 3\hbar$ ($= I_+ + 3\hbar$) if the phonon is aligned with the rotation axis and $\Delta i_x = 3\hbar$; see figure 1. What is evident in figure 2 is that for some nuclei, such as $^{222,224,226}$Ra, $^{224,226}$Th and $^{226}$U, $\Delta i_x$ approaches zero for increasing values of $\omega$, where pairing effects become smaller. These isotopes of radium, thorium and uranium are good candidates for having rigid pear shapes. By contrast, the radon isotopes and the more neutron-rich isotopes of Ra, Th and U appear to be octupole vibrational, as are most nuclei with $Z \approx 56$, $N \approx 88$ (figure 3), for which $\Delta i_x \approx 3\hbar$.

For odd-mass nuclei, the odd proton or neutron has angular momentum $j$ so that the total angular momentum $I = R + j$ has projection $K = j_z$ on the $z$-axis. In this case, there will be two bands containing states of alternating parity,

$$I^{\pi} = K^{\pm}, (K+1)^{\mp}, (K+2)^{\pm}, (K+3)^{\mp}, \ldots . \tag{2.1}$$

The parities of the states $\pi$ are $(-)^{I+(1/2)}$ in one band and $(-)^{I-(1/2)}$ in the other. The two bands lie close in energy and form a *parity doublet* such as has been observed in $^{225}$Ra [22] and $^{223}$Th [23]. A discussion of the experimental evidence for parity doublets, which are relevant to atomic electric-dipole moment (EDM) searches (see later), is given in [8–11].

## 3. Experimental evidence: electric charge distribution

Convincing evidence that nuclei have quadrupole deformation comes from the observation of large electric quadrupole ($E$2) moments arising from the charge distribution in the deformed shape. In a similar way, octupole shapes will give rise to enhanced electric dipole ($E$1) and octupole ($E$3) moments for nuclear transitions between states of opposite parity. Enhanced $E$1 moments arise, in a simple picture, from the lightning rod effect where the charge accumulates on the more pointed end of the pear shape, causing a separation of the centre of charge and the centre of mass. Although this can increase the $E$1 moment by several orders of magnitude, there will be large fluctuations in its value because the interacting nucleons contribute individually and collectively [24,25]. This can give rise to a net moment of nearly zero, as has been observed in $^{146}$Ba [14] and $^{224}$Ra [26]. A more reliable indicator of octupole correlations is the $E$3 moment that arises from the reflection-asymmetric charge distribution throughout the nuclear volume, which largely depends on the collective behaviour of the nucleus. As will be seen later, although both octupole vibrations and octupole deformation will give rise to large $E$3 moments, there are now sufficient experimental data to distinguish between the two types of instability.

In order to determine $E$3 transition moments between the nuclear states the method of Coulomb excitation is usually employed. Here the states are excited by the electromagnetic interaction between heavy ions at relative energies just below the Coulomb barrier. Whereas $E$1, $E$2 and magnetic dipole ($M$1) transitions dominate in the electromagnetic decay of nuclear states, $E$2 and $E$3 transitions dominate the Coulomb excitation process. This allows the $E$2 and $E$3 moments to be determined from measurement of the yields of the states populated in the reaction, often inferred from measurement of the yields of the $\gamma$-rays de-exciting these states [27]. Following a proof-of-principle experiment performed at the Nuclear Structure Facility, Daresbury Laboratory, UK [28], extensive measurements were made of $E$3 matrix elements between states in the ground-state (positive-parity) rotational band and the octupole (negative-parity) band in the stable isotope $^{148}$Nd, using a number of targets and beams provided by facilities at Daresbury, the University of Rochester, NY, USA, and the Oak Ridge National Laboratory, TN, USA [29,30]. Similar measurements were carried out at the Munich Tandem Laboratory, Germany, and GSI Darmstadt, Germany, for $^{226}$Ra [31], whose half-life (1600 yr) is sufficiently long for use as target material. For 20 years, these were the only nuclei in the two regions of strong octupole collectivity where a comprehensive knowledge of $E$3 matrix elements existed. To explore further required the development of accelerated radioactive ion beams (RIBs), which enables a much larger range of isotopes to be studied using the method of Coulomb excitation. In this manner, measurements of $E$3 matrix elements have been performed in recent years for $^{220}$Rn [32], $^{224}$Ra [32], $^{144}$Ba [33], $^{146}$Ba [34] and $^{222,228}$Ra [35], using the RIB accelerators REX- and HIE-ISOLDE, CERN, Geneva, Switzerland, and CARIBU, Argonne National Laboratory, Lemont, IL, USA. (It should be noted that the measurements [7,17] of energy levels of $^{224,226}$Rn presented in figure 2 also exploited Coulomb excitation of accelerated radon ions, but at higher energies where the interaction is not purely electromagnetic.)

The measured quadrupole moment $\mathcal{Q}_2$ and octupole moment $\mathcal{Q}_3$ for nuclei in the $Z \approx 88$, $N \approx 134$ mass region are compared with those measured for adjacent heavy nuclei in figure 4. In this figure, the intrinsic moments $\mathcal{Q}_{\lambda}$ are derived [35] from the transition matrix elements

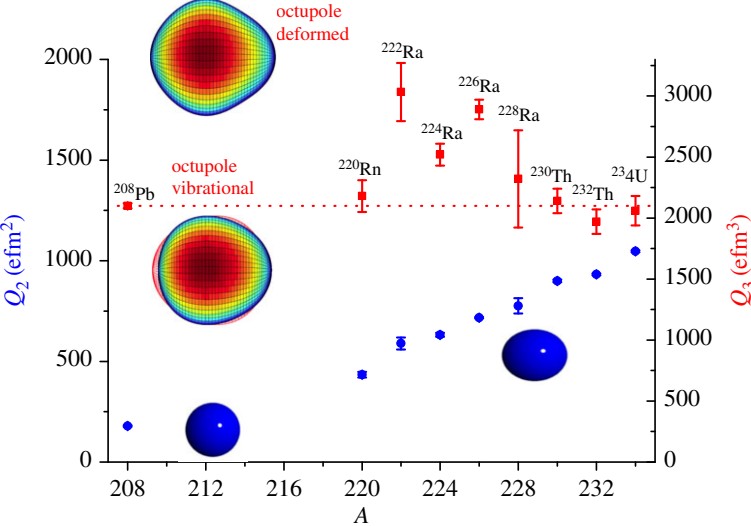

**Figure 4.** The systematics of measured $E2$ and $E3$ intrinsic moments $Q_\lambda$ for $0^+ \rightarrow 2^+$ and $0^+ \rightarrow 3^-$ transitions, respectively, in the heavy mass region ($A \geq 208$). For the source of the data shown here, see fig. 14 in [12], and see [35]. (Online version in colour.)

$\langle I_i || \mathcal{M}(E\lambda) || I_f \rangle$ corresponding to the $0^+ \rightarrow 2^+$ ($E2$) and $0^+ \rightarrow 3^-$ ($E3$) transitions, assuming the validity of the rotational model. It is striking that, while the value of $\mathcal{Q}_2$ increases by a factor of 6 between $^{208}$Pb and $^{234}$U, the value of $\mathcal{Q}_3$ changes by only 50% in the entire mass region. Nevertheless, the larger $Q_3$ values for $^{222}$Ra, $^{224}$Ra and $^{226}$Ra indicate an enhancement in octupole collectivity that is consistent with an onset of octupole deformation in this mass region. It is also observed that $\mathcal{Q}_3$ is lower for $^{224}$Ra than for $^{222}$Ra and $^{226}$Ra. It remains to be seen whether this dip arises from an experimental artefact or there is a real physical effect. The octupole moments for nuclei in the $Z \approx 56$, $N \approx 88$ mass region are shown in figure 5. Here it is not clear whether a maximum in the value of $\mathcal{Q}_3$ occurs at $N \approx 88$; more precise measurements for $^{152}$Gd and $^{144,146}$Ba would be desirable. For $A > 226$ (figure 4) and for $N > 88$ (figure 5), a reduction in the value of $\langle 0^+ || \mathcal{M}(E3) || 3^- \rangle$ could arise because the $E3$ strength is no longer concentrated in the lowest octupole band but is shared among this band and bands with other modes of octupole shape oscillations that occur in deformed nuclei. These other modes will come down in energy as the number of protons and neutrons move away from the closed shell at $Z = 50$, $N = 82$ or at $Z = 82$, $N = 126$. The manner in which the $E3$ strength is distributed among different octupole bands has been studied in a few cases in the lighter mass region (e.g. [36]).

A summary of the values of $\mathcal{Q}_3$ derived from the measured $E3$ matrix elements for different transitions in the Ba–Nd and Rn–Ra octupole regions is given in figure 6. The data shown in this figure are restricted to measurements employing multi-step Coulomb excitation. In the figure, the values of $\mathcal{Q}_3$ are shown separately for transitions $I^+ \rightarrow (I+1)^-$, $I^+ \rightarrow (I+3)^-$, $I^- \rightarrow (I+1)^+$ and $I^- \rightarrow (I+3)^+$. It is observed that the values of $\mathcal{Q}_3$ for all transitions in $^{222,224,226}$Ra are approximately constant, consistent with the picture of a rotating pear shape. The behaviour of the energy levels (figure 2) together with the enhancement (figure 4) and rotor-like pattern of the electric octupole moments $\mathcal{Q}_3$ (figure 6) provide compelling evidence that these isotopes of radium have stable octupole deformation, and so far these are the only cases in nature where such evidence exists. By contrast, the values of $\mathcal{Q}_3$ corresponding to the $2^+ \rightarrow 3^-$ and $1^- \rightarrow 4^+$ transitions in $^{148}$Nd and $^{228}$Ra are close to zero, while for $^{144,146}$Ba and $^{220}$Rn there are only measurements for the $I^+ \rightarrow (I+3)^-$ transitions. For these nuclei the available data on the $E3$ moments do not yet support a description in terms of rigid pear shapes, and the behaviour of the energy levels is consistent with that expected for octupole vibrators.

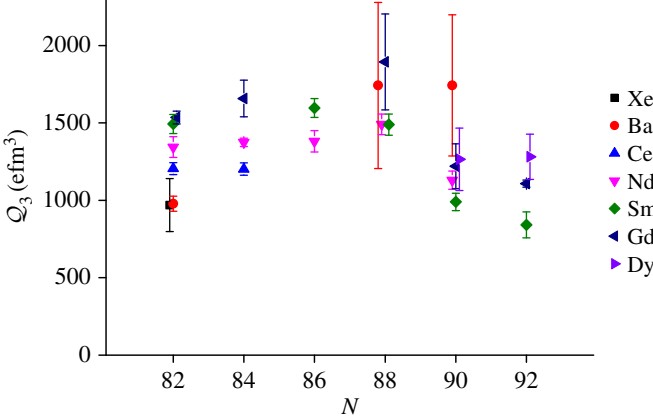

**Figure 5.** The systematics of measured *E*3 intrinsic moments, $\mathcal{Q}_3$, for $0^+ \rightarrow 3^-$ transitions in the $Z \approx 56$, $N \approx 88$ mass region. For the source of the data shown here, see fig. 15 in [12], and see [34]. (Online version in colour.)

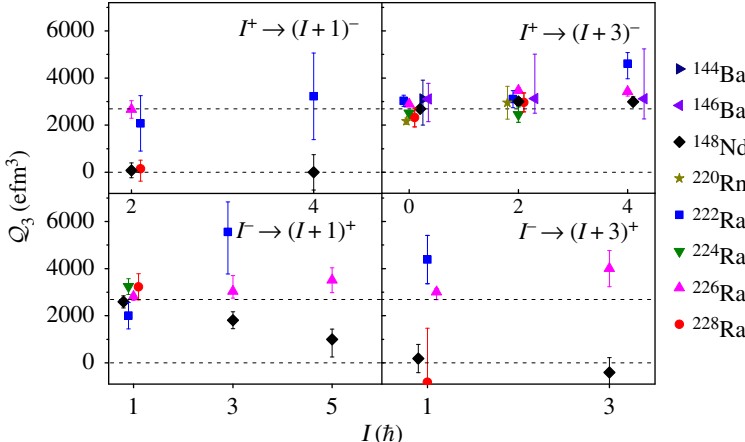

**Figure 6.** Values of the intrinsic octupole moments, $\mathcal{Q}_3$, for various transitions in nuclei where octupole correlations are the strongest. Here the values of $\mathcal{Q}_3$ are shown separately for transitions connecting $I^+ \rightarrow (I+1)^-$, $I^+ \rightarrow (I+3)^-$, $I^- \rightarrow (I+1)^+$ and $I^- \rightarrow (I+3)^+$. The upper dashed line is the average value of $\mathcal{Q}_3(0^+ \rightarrow 3^-)$ for the radium isotopes. To aid comparison the values of $\mathcal{Q}_3$ for $^{144}$Ba, $^{146}$Ba and $^{148}$Nd have been multiplied by 1.78. The data are taken from [30–35]. (Online version in colour.)

The values of intrinsic octupole moments, $Q_3$, for radium isotopes, deduced from the measured transition matrix element $\langle 0^+ || \mathcal{M}(E3) || 3^- \rangle$, are compared with various theoretical calculations in figure 7. The calculations are from macroscopic–microscopic (MacMic) [37], relativistic mean-field (RMF) (NL1 variant) [38], cluster model [39], Gogny Hartree–Fock–Bogoliubov (Gogny) (D1S variant) [40], relativistic Hartree–Bogoliubov + interacting boson model (RHIBM) [41], quadrupole–octupole collective Hamiltonian (QOCH) [42] and Skyrme Hartree–Fock–Bogoliubov (Skyrme) (UNEDF0 variant) [43] calculations. All of these models except for the cluster model predict a maximum around $N = 136$–138. The variation in the predicted values from the different theories is evident, but no particular model description can be favoured or discarded on the basis of the experimental data. Measurements for $^{220}$Ra ($N = 132$) and $^{230}$Ra ($N = 142$) would be desirable, as well as increased precision for the $^{228}$Ra ($N = 140$) measurement and confirmation of the minimum observed at $N = 136$.

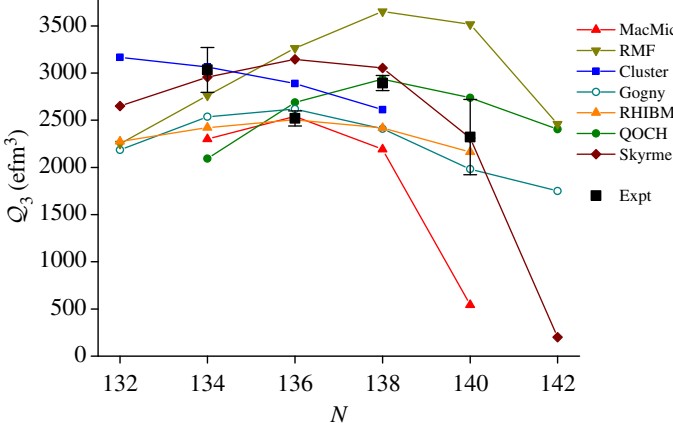

**Figure 7.** Measured values [31,32,35] (Expt) of intrinsic octupole moments, $Q_3$, for $0^+ \rightarrow 3^-$ transitions as a function of $N$ for radium isotopes, compared with values calculated using various theoretical models. The QOCH value for $^{230}$Ra is from Z.P. Li (2020, personal communication). For details of the other calculations see the text. (Online version in colour.)

## 4. Pear shapes and electric-dipole moments

Atoms with octupole-deformed nuclei are very important in the search for permanent atomic electric-dipole moments (EDMs). Consider the time-reversal transformation ($\mathcal{T}$) for which $t \rightarrow -t$. Since the orientation of a system can be specified only by the orientation of its angular momentum, the electric dipole moment $\boldsymbol{d}$ and the angular momentum $\boldsymbol{I}$ must transform their signs in the same way under $\mathcal{T}$. However, $\boldsymbol{I}$ changes sign under $\mathcal{T}$ but $\boldsymbol{d}$ does not, so $\boldsymbol{d}$ must vanish if there is ($\mathcal{T}$) symmetry [44]. (Similarly, $\boldsymbol{d}$ must vanish if there is symmetry under a parity transformation, $\mathcal{P}$, where $\boldsymbol{r} \rightarrow -\boldsymbol{r}$.) Under the assumption of the $\mathcal{CPT}$ theorem, if $\mathcal{T}$ is violated then $\mathcal{CP}$ must also be violated, and the observation of a substantial non-zero EDM would indicate $\mathcal{CP}$ violation owing to physics beyond the standard model (SM). Such 'flavour-diagonal' $\mathcal{CP}$ violation has not yet been observed, although modifications of the SM that would give rise to measurable effects are strongly motivated in order to account for the observed cosmological dominance of baryons over antibaryons. In fact, experimental limits on EDMs provide important constraints on many proposed extensions to the standard model [45,46]. For a review of this topic, see [47].

For a neutral atom in its ground state the Schiff moment, the electric-dipole distribution weighted by radius squared, is the lowest-order observable nuclear moment. A $\mathcal{CP}$-violating Schiff moment will induce a major contribution to the atomic EDM. Odd-$A$ octupole-deformed nuclei will have enhanced nuclear Schiff moments owing to the presence of the large octupole collectivity and nearly degenerate parity doublets, both of which will occur in pear-shaped nuclei [48–51]. The sensitivity to $\mathcal{CP}$ violation of an EDM measurement in an atom with a pear-shaped nucleus can in principle be improved by a large factor [50] compared with that for a non-octupole-enhanced system such as $^{199}$Hg, currently providing the most stringent limit for diamagnetic atoms [46]. Experimental programmes are in place to measure EDMs in atoms of odd-$A$ Rn and Ra isotopes in the octupole region (e.g. [52,53]). Essential in the interpretation of the observational limits is a detailed understanding of the structure of these nuclei [54]. The recent measurements described in the previous section conclude that the even–even nuclei $^{222-226}$Ra have octupole-deformed character, and their odd-mass neighbours $^{223,225}$Ra, having parity doublets separated by $\approx 50\,\text{keV}$ [11], should have large enhancement of their Schiff moments. Measurements of the $E3$ strength in odd-$A$ nuclei have yet to be carried out, however. For radon isotopes, it appears unlikely that odd-$A$ nuclei such as $^{223,225}$Rn will have low-lying parity doublets [7,17], and any enhancement of the Schiff moment will be smaller in radon atoms

than for radium atoms. Realistic estimates of Schiff moments for octupole-vibrational systems have yet to be made [55,56].

## 5. Summary and outlook

There is now a substantial body of evidence, from the behaviour of the energies of quantum states and the interconnecting electromagnetic matrix elements, particularly electric octupole matrix elements, that a few isotopes of radium have permanent octupole deformation, i.e. are pear shaped. This is important not just for testing nuclear theories but also for improving the sensitivity of atomic EDM searches that could reveal the violation of fundamental symmetries not accounted for by the standard model. The systematic behaviour of energy levels in certain isotopes of thorium and uranium nuclei suggests that these may also be pear shaped, and calculations using the Gogny Hartree–Fock–Bogoliubov method [57] predict very large values of $E3$ moments in Th, U and Pu isotopes with $N \approx 134$ (see fig. 19 in [12]) . Experiments to measure $E3$ transition probabilities in these heavier nuclei, and in odd-mass nuclei relevant to EDM searches, await advances in radioactive beam technology that should be realized in the next few years.

Data accessibility.  This article has no additional data.

Competing interests.  I declare I have no competing interest.

Funding.  This work was supported by the Science and Technology Facilities Council (UK) grant no. ST/P004598/1.

Acknowledgements.  I express my appreciation to Rafał Broda, Yuchen Cao, Tim Chupp, Doug Cline, James Cocks, Tomek Czosnyka (dec.), Giacomo de Angelis, Jacek Dobaczewski, Bogdan Fornal, Liam Gaffney, Paul Greenlees, Mark Huyse, David Jenkins, Graham Jones, David Joss, Rauno Julin, Thorsten Kröll, Matti Leino, Zhipan Li, Bondili Nara Singh, Witek Nazarewicz, Kosuke Nomura, Robert Page, Luis Robledo, Marcus Scheck, Timur Shneidman, John Smith, Pietro Spagnoletti, Frank Stephens, Piet Van Duppen, Nigel Warr, Fredrik Wenander, Ching-Yen Wu, Magda Zielińska and many others with whom it has been a privilege to collaborate.

## Author profile

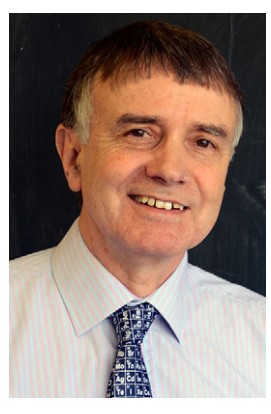

Peter Butler carried out his undergraduate studies at King's College London and received a PhD from the University of Liverpool under the supervision of John Sharpey-Schafer. After 3 years of postdoctoral research at the Lawrence Berkeley Laboratory he returned to Liverpool in 1978, receiving an Advanced Fellowship from the Science Research Council. Since then he has spent sabbaticals at the Oak Ridge National Laboratory and at the University of Jyväskylä. For the period 2002–2005 he was seconded to CERN to lead the ISOLDE group, followed by roles of spokesperson and chair of the ISOLDE collaboration.

Peter's research has focused on experimental studies of the phenomenon of reflection asymmetry in atomic nuclei and the structure of the heaviest elements. He presented compelling evidence that some nuclei have permanent octupole deformation ('pear shape') from measurements of their rotational behaviour and electric charge distribution. He also conceived and led the development of instruments at Jyväskylä to measure quantum transitions in heavy nuclei: the GREAT spectrometer that identifies transitions between quantum states in rare nuclei, and SACRED that measures atomic conversion electrons emitted by high-Z nuclei. In his association with CERN he has been strongly involved with the HIE-ISOLDE project and has more recently been actively involved with the STFC project to construct a solenoidal spectrometer at ISOLDE.

Peter was appointed to a Personal Chair in the Department of Physics at the University of Liverpool in 1999. In 2012, he was awarded the Rutherford Medal of the Institute of Physics for his contributions to experimental nuclear physics. In 2019 he was awarded an Honorary Doctorate

in Philosophy by the University of Jyväskylä, and in the same year elected a Fellow of the Royal Society.

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
