## [Reviewer comments · Proceedings. Mathematical, Physical, and Engineering Sciences]

Review History

RSPA-2020-0202.R0 (Original submission)

Review form: Referee 1

Is the manuscript an original and important contribution to its field?

Excellent

Is the paper of sufficient general interest?

Excellent

Is the overall quality of the paper suitable?

Excellent

Can the paper be shortened without overall detriment to the main message?

Yes

Do you think some of the material would be more appropriate as an electronic appendix?

No

Do you have any ethical concerns with this paper?

No

Recommendation?

Accept with minor revision (please list in comments)

Comments to the Author(s)

The manuscript is well written update and summary of octupole deformation in medium heavy and heavy nuclei. In addition to addressing the relevant points with respect to nuclear structure physics also potential role of these nuclei in the searches for permanent electric dipole moment in atoms are addressed. The author has played central role in recent studies of this phenomena using Coulomb excitation of radioactive ion beams.

I only have only two comments on the text itself:

page 5, line 5: will be large

page 6, line 45: instead of using the word "statistical fluctuation" I would suggest to use "statistical uncertainty"

Review form: Referee 2

Is the manuscript an original and important contribution to its field?

Excellent

Is the paper of sufficient general interest?

Excellent

Is the overall quality of the paper suitable?

Excellent

Can the paper be shortened without overall detriment to the main message?

Yes

Do you think some of the material would be more appropriate as an electronic appendix?

No

Do you have any ethical concerns with this paper?

No

Recommendation?

Accept with minor revision (please list in comments)

Comments to the Author(s)

This is an excellent review article, and I recommend its publication in Proceedings A.

I have only minor and rather technical suggestions.

In page 6, line 5, "will large fluctuations ..." seems to be an incomplete sentence.

In Fig. 1, 3 ρ is not visible.

In Fig. 2, axes may help, and some letters are too small.

In Fig. 3. some letters are too small.

In Fig. 4, Q₂, Q₃ and A are too large out of balance.

In Fig. 6, there are four panels, which is not clear. They should be divided. The legend is common to all panels, but this is not obvious.

Caption of Fig. 6, line 3, "The upper dashed line" does not exist in the figure.

Fig. 7, letters are large enough. This figure is a good model to the other figures.

Review form: Referee 3

Is the manuscript an original and important contribution to its field?

Excellent

Is the paper of sufficient general interest?

Excellent

Is the overall quality of the paper suitable?

Excellent

Can the paper be shortened without overall detriment to the main message?

Yes

Do you think some of the material would be more appropriate as an electronic appendix?

No

Do you have any ethical concerns with this paper?

No

Recommendation?

Accept with minor revision (please list in comments)

Comments to the Author(s)

This paper presents the present status of experimental evidence for the occurrence of octupole shapes in atomic nuclei and reviews the behaviour of energy levels and electric octupole transition moments.

The work is thoroughly implemented, the manuscript is well written and the subject is of a high actuality. Therefore, I would recommend for publication after the author provide a clarification of the following point.

1) In lines 30,31 of page 5, there is a sentence "Deviations from this ideal behaviour will occur for low-lying states because of the dominant effect of nucleonic pairing." Could the author say some more words to explain this comment or cite some relevant references?

Review form: Referee 4

Is the manuscript an original and important contribution to its field?

Excellent

Is the paper of sufficient general interest?

Good

Is the overall quality of the paper suitable?

Excellent

Can the paper be shortened without overall detriment to the main message?

Yes

Do you think some of the material would be more appropriate as an electronic appendix?

No

Do you have any ethical concerns with this paper?

No

Recommendation?

Accept as is

Comments to the Author(s)

The manuscript is very compact yet informative (I've learned that Rn appears to be less suitable to the EDM measurement than Ra). I do not have any suggestions for its improvement.

Perhaps a word in "defence" of Hg EDM experiment: while of course the octupole-shaped deformations provide a way of enhancing the Schiff moment, the final utility of the bound/measurement comes down to the size of the errors on EDMs. ^{199}Hg is a stable isotope that can be used in copious amounts in the EDM measurement, while suitable Ra exists in tiny quantities - which will lead to a constraint on atomic Ra EDM to be less stringent than in Hg simply because the statistical errors in Hg EDM are the smallest across all EDM experiments. It remains to be seen whether the deformation-induced enhancement of the effects in Ra would in the future make it competitive with, or better than Hg.

Decision letter (RSPA-2020-0202.R0)

08-May-2020

Dear Professor Butler,

On behalf of the Reviews Editor, I am pleased to inform you that your Manuscript RSPA-2020-0202 entitled "Pear-shaped atomic nuclei" has been accepted for publication subject to minor revisions in Proceedings A. Please find the referees' comments below. We were fortunate in that all the invited referees accepted and returned their reviews promptly, a tribute to you and the excellence of your paper. Thank you.

The reviewer(s) have recommended publication, but also suggest some minor revisions to your manuscript. Therefore, I invite you to respond to the reviewer(s)' comments and revise your manuscript. It is a condition of publication that you submit the revised version of your manuscript within 7 days. If you do not think you will be able to meet this date please let me know in advance of the due date. A revised paper will not be returned to the referees for further review.

To revise your manuscript, log into <https://mc.manuscriptcentral.com/prsa> and enter your Author Centre, where you will find your manuscript title listed under "Manuscripts with Decisions." Under "Actions," click on "Create a Revision." Your manuscript number has been appended to denote a revision.

You will be unable to make your revisions on the originally submitted version of the manuscript. Instead, revise your manuscript and upload a new version through your Author Centre.

IMPORTANT: Your original files are available to you when you upload your revised manuscript. Please delete any redundant files before completing the submission process.

In addition to addressing all of the reviewers' and editor's comments, your revised manuscript **MUST** contain the following sections before the reference list (for any heading that does not apply to your work, please include a comment to this effect):

- Acknowledgements
- Funding statement

See <https://royalsociety.org/journals/authors/author-guidelines/> for further details.

When uploading your revised files, please make sure that you include the following as we cannot proceed without these:

- 1) A text file of the manuscript (doc, txt, rtf or tex), including the references, tables (including captions) and figure captions. Please remove any tracked changes from the text before submission. PDF files are not an accepted format for the "Main Document".
- 2) A separate electronic file of each figure (tif, eps or print-quality pdf preferred). The format should be produced directly from original creation package, or original software format.
- 3) Electronic Supplementary Material (ESM): all supplementary materials accompanying an accepted article will be treated as in their final form. Note that the Royal Society will not edit or typeset supplementary material and it will be hosted as provided. Please ensure that the supplementary material includes the paper details where possible (authors, article title, journal name). Supplementary files will be published alongside the paper on the journal website and posted on the online figshare repository (<https://figshare.com>). The heading and legend provided for each supplementary file during the submission process will be used to create the figshare page, so please ensure these are accurate and informative so that your files can be found in searches. Files on figshare will be made available approximately one week before the accompanying article so that the supplementary material can be attributed a unique DOI. Alternatively you may upload a zip folder containing all source files for your manuscript as described above with a PDF as your "Main Document". This should be the full paper as it appears when compiled from the individual files supplied in the zip folder.

Article Funder

Please ensure you fill in the Article Funder question on page 2 to ensure the correct data is collected for FundRef (<http://www.crossref.org/fundref/>).

Media summary

Please ensure you include a short non-technical summary (up to 100 words) of the key findings/importance of your paper. This will be used for to promote your work and marketing purposes (e.g. press releases). The summary should be prepared using the following guidelines:

- *Write simple English: this is intended for the general public. Please explain any essential technical terms in a short and simple manner.
- *Describe (a) the study (b) its key findings and (c) its implications.
- *State why this work is newsworthy, be concise and do not overstate (true 'breakthroughs' are a rarity).
- *Ensure that you include valid contact details for the lead author (institutional address, email address, telephone number).

Cover images

We welcome submissions of images for possible use on the cover of Proceedings A. Images should be square in dimension and please ensure that you obtain all relevant copyright permissions before submitting the image to us. If you would like to submit an image for consideration please send your image to proceedingsa@royalsociety.org

Once again, thank you for submitting your manuscript to Proceedings A and I look forward to receiving your revision. If you have any questions at all, please do not hesitate to get in touch.

Best wishes
 Raminder Shergill
proceedingsa@royalsociety.org
 Proceedings A

on behalf of
 Professor Chris Garrett
 Reviews Editor
 Proceedings A

Reviewer(s)' Comments to Author:

Referee: 1

Comments to the Author(s)

The manuscript is well written update and summary of octupole deformation in medium heavy and heavy nuclei. In addition to addressing the relevant points with respect to nuclear structure physics also potential role of these nuclei in the searches for permanent electric dipole moment in atoms are addressed. The author has played central role in recent studies of this phenomena using Coulomb excitation of radioactive ion beams.

I only have only two comments on the text itself:

page 5, line 5: will be large

page 6, line 45: instead of using the word "statistical fluctuation" I would suggest to use "statistical uncertainty"

Referee: 2

Comments to the Author(s)

This is an excellent review article, and I recommend its publication in Proceedings A.

I have only minor and rather technical suggestions.

In page 6, line 5, "will large fluctuations ..." seems to be an incomplete sentence.

In Fig. 1, 3 μ hoar is not visible.

In Fig. 2, axes may help, and some letters are too small.

In Fig. 3. some letters are too small.

In Fig. 4, Q₂, Q₃ and A are too large out of balance.

In Fig. 6, there are four panels, which is not clear. They should be divided. The legend is common to all panels, but this is not obvious.

Caption of Fig. 6, line 3, "The upper dashed line" does not exist in the figure.

Fig. 7, letters are large enough. This figure is a good model to the other figures.

Takaharu Otsuka

Referee: 3

Comments to the Author(s)

This paper presents the present status of experimental evidence for the occurrence of octupole shapes in atomic nuclei and reviews the behaviour of energy levels and electric octupole transition moments.

The work is thoroughly implemented, the manuscript is well written and the subject is of a high actuality. Therefore, I would recommend for publication after the author provide a clarification of the following point.

1) In lines 30,31 of page 5, there is a sentence "Deviations from this ideal behaviour will occur for low-lying states because of the dominant effect of nucleonic pairing." Could the author say some more words to explain this comment or cite some relevant references?

Referee: 4

Comments to the Author(s)

The manuscript is very compact yet informative (I've learned that Rn appears to be less suitable to the EDM measurement than Ra). I do not have any suggestions for its improvement.

Perhaps a word in "defence" of Hg EDM experiment: while of course the octupole-shaped deformations provide a way of enhancing the Schiff moment, the final utility of the bound/measurement comes down to the size of the errors on EDMs. ^{199}Hg is a stable isotope that can be used in copious amounts in the EDM measurement, while suitable Ra exists in tiny quantities - which will lead to a constraint on atomic Ra EDM to be less stringent than in Hg simply because the statistical errors in Hg EDM are the smallest across all EDM experiments. It remains to be seen whether the deformation-induced enhancement of the effects in Ra would in the future make it competitive with, or better than Hg.

Maxim Pospelov

Author's Response to Decision Letter for (RSPA-2020-0202.R0)

See Appendix A.

Decision letter (RSPA-2020-0202.R1)

18-May-2020

Dear Professor Butler

On behalf of the Editor, I am pleased to inform you that your manuscript entitled "Pear-shaped atomic nuclei" has been accepted in its final form for publication in Proceedings A.

Our Production Office will be in contact with you in due course. You can expect to receive a proof of your article soon. Please contact the office to let us know if you are likely to be away from e-mail in the near future. If you do not notify us and comments are not received within 5 days of sending the proof, we may publish the paper as it stands.

Under the terms of our licence to publish you may post the author generated postprint (ie. your accepted version not the final typeset version) of your manuscript at any time and this can be made freely available. Postprints can be deposited on a personal or institutional website, or a recognised server/repository. Please note however, that the reporting of postprints is subject to a media embargo, and that the status the manuscript should be made clear. Upon publication of the definitive version on the publisher's site, full details and a link should be added.

You can cite the article in advance of publication using its DOI. The DOI will take the form: 10.1098/rspa.XXXX.YYYY, where XXXX and YYYY are the last 8 digits of your manuscript number (eg. if your manuscript number is RSPA-2017-1234 the DOI would be 10.1098/rspa.2017.1234).

For tips on promoting your accepted paper see our blog post:
<https://blogs.royalsociety.org/publishing/promoting-your-latest-paper-and-tracking-your-results/>

Thank you for your submission. On behalf of the Editors of the journal, we look forward to your continued contributions to the Journal.

Best wishes

Raminder Shergill,
Proceedings A Editorial Office
proceedingsa@royalsociety.org

Appendix A

I thank all referees for their careful reading of the paper, and have made the following changes to address their comments:

Referee 1

I have corrected the phrase in section 3 to read "...there will be large fluctuations in its value..."

For the other phrase I have replaced it by "... whether this dip arises from an experimental artifact or there is a real physical effect."

Referee 2 (Prof. Otsuka)

I have corrected the phrase in section 3, see above.

Figure 1 is taken from Butler et al. 2019; the "3 hbar" label should be visible.

In Figures 2, 3, 4, I have adjusted the sizes of the legends and axis labels where necessary.

In Figure 6 I have inserted boundary lines for each of the panels and have modified the legend. The upper dashed line, at a value of 2700 efm^3 , should be visible.

Referee 3

I have expanded this sentence (section 2) to read:

"Deviations from this ideal behaviour will occur for low-spin states where nucleonic pairing will try to maintain a reflection-symmetric shape" and inserted a reference to Egido JL, Robledo LM 1989 (Microscopic study of the octupole degree of freedom in the radium and thorium isotopes with Gogny forces. Nucl. Phys. A 494, 85-101).

Referee 4 (Prof. Pospelov)

I agree with the comment regarding how competitive EDM measurements in unstable ^{225}Ra will be compared to those in stable ^{199}Hg . It is clear that the ^{225}Ra experimental programme at Argonne (and FRIB) requires possibly decades of development to reach the sensitivity achieved by the Seattle group. However, even if only the same sensitivity is reached, it is (apparently) more straightforward to calculate the Schiff moment and atomic EDM for the deformed Ra system than for the near-spherical Hg system.

I have re-worded the sentence in section 4 to read "The sensitivity to CP violation of an EDM measurement in an atom with a pear-shaped nucleus can in principle be improved by a large factor..." to reflect the challenges for ^{225}Ra and also make the meaning clearer.

In addition I have made the following clarifications, corrections and additions:

Introduction

I have added "in nuclei" in "For a review of generalised spontaneous symmetry breaking in nuclei...".

Section 3

I have replaced the theoretical calculations "Skyrme Hartree-Fock (SkO' variant)" (Engel et al., 2003) by "Skyrme Hartree-Fock-Bogoliubov (UNEDF0 variant)" (Cao et al., 2020). The former publication did not contain explicit calculations of E3 moments; these I estimated from the shape parameters. The latter paper, made available to me recently, presents E3 moments properly calculated from the octupole charge distribution.

Figures

Figure 1: the caption now refers only to Butler PA et al. 2019.

Figure 2: I have repaired the Th lines for N=132 and N=140.

Figure 7: I have replaced the Skyrme calculations of Engel et al. by those of Cao et al.

References

I have replaced "Butler PA et al. 2020 The observation of vibrating pear-shapes in radon nuclei: update. <http://arxiv.org/abs/2003.10147>." by "Butler PA et al. 2020 Addendum: The observation of vibrating pear-shapes in radon nuclei. Nat. Commun. to be published". This is now the primary reference for this work.

I have replaced "Engel J, Bender M, Dobaczewski J, de Jesus JH, Olbratowski P 2003 Time-reversal violating Schiff moment of ^{225}Ra . Phys. Rev. C 68, 025501." by "Cao Y, Agbemava SE, Afanasjev AV, Nazarewicz W, Olsen E. 2020 Landscape of pear-shape even-even nuclei. <http://arxiv.org/abs/2004.01319v1>".

References [7], [17]-[20] are now placed in the correct order.